# A last-in first-out stack data structure implemented in DNA

Annunziata Lopiccolo [1,4], Ben Shirt-Ediss [1,4], Emanuela Torelli [1], Abimbola Feyisara Adedeji Olulana [2,3], Matteo Castronovo [2,3], Harold Fellermann [1✉] & Natalio Krasnogor [1✉]

DNA-based memory systems are being reported with increasing frequency. However, dynamic DNA data structures able to store and recall information in an ordered way, and able to be interfaced with external nucleic acid computing circuits, have so far received little attention. Here we present an in vitro implementation of a stack data structure using DNA polymers. The stack is able to record combinations of two different DNA signals, release the signals into solution in reverse order, and then re-record. We explore the accuracy limits of the stack data structure through a stochastic rule-based model of the underlying polymerisation chemistry. We derive how the performance of the stack increases with the efficiency of washing steps between successive reaction stages, and report how stack performance depends on the history of stack operations under inefficient washing. Finally, we discuss refinements to improve molecular synchronisation and future open problems in implementing an autonomous chemical data structure.

[1] Interdisciplinary Computing and Complex Biosystems Research Group, School of Computing, Newcastle University, Newcastle-upon-Tyne, UK. [2] Food Colloids and Bioprocessing Group, School of Food Science and Nutrition, University of Leeds, Leeds, UK. [3] Regional Referral Centre for Rare Diseases, Azienda Sanitaria Universitaria Integrata di Udine, Udine, Italy. [4] These authors contributed equally: Annunziata Lopiccolo, Ben Shirt-Ediss. ✉email: harold.fellermann@newcastle.ac.uk; natalio.krasnogor@newcastle.ac.uk

DNA is being used to engineer an increasing array of biochemical memory devices, both in vitro and in vivo[1]. To date, three broad classes of DNA-based memory platforms can be identified: bit memory, archival memory, and associative memory.

Bit memory platforms perform the setting (and sometimes resetting) of individual bits of information when signals are transiently present. In vivo, DNA bit memories have traditionally been engineered as volatile multistable genetic networks[1,2]. More recently, non-volatile in vivo approaches have also been realised that edit single DNA nucleotides (or longer sections of bases) on cellular genomes or plasmids via recombinase or CRISPR techniques[3]. Such in vivo bit editing is billed as a future technology for the in situ recording of intracellular events, persistent across cell generations. Devices such as logic gates that remember past inputs[4] or sequential logic sensitive to the order of input signals[5,6] demonstrate that in vivo bit editing approaches can additionally intertwine memory with basic computation. In vitro, a volatile single bit toggle-on/-toggle-off memory has been demonstrated, based on an away-from-equilibrium chemical reaction network of DNA templates and three enzymes[7]. Conversely, non-volatile DNA memory has been constructed in the test tube by utilising the hybridisation of complementary DNA strands as the basis of addressable memory bits. Writing and erasure of bits has been achieved via the use of temperature changes[8], isothermal strand displacement[9,10], and electric fields to co-localise strands[11]. In a second paradigm, archival memory platforms use chemically synthesised DNA as a high density long-term data storage medium; information (e.g., images or text) is encoded into a base sequence, to be later read out via sequencing or PCR (reviewed in refs. [12,13]). Finally, associative memory platforms indicate whether a new or incompletely presented pattern is close to a previously remembered pattern. Existing DNA in vitro implementations have been based on hybridisation and amplification[14] and more recently on a strand displacement network that emulates a 4-input Hopfield network[15].

A fourth potential class of memory platform exists, but has been little explored in DNA: that of a data structure. In Computer Science, a data structure is an object that not only stores information but also organises that information in a way that facilitates the efficient querying, searching, modification, and removal of the data with respect to a particular application context[16]. Information in a data structure is manipulated through a series of functions (specified by its Abstract Data Type signature), which preserve the organisation of the data. The efficient information layouts that data structures achieve make them the cornerstones of complex computational algorithms. However, to date, biochemical realisations of data structures are lacking.

In this study, we experimentally implement and computationally model a stack data structure (Fig. 1a–c) as a DNA chemical reaction system (Fig. 1d). The stack stores and retrieves information (DNA signal strands) in a last-in first-out order by building and truncating DNA "polymers" of single stranded DNA (ssDNA) strands. Ultimately, our stack data structure is intended for use in a dynamic DNA computing context[17], where interfacing nucleic acid circuits in the same solution are able to use the functions provided by the data structure (Fig. 1b) to store signal strands in one order, and then release them in the opposite order at a later time. Once released, a previously stored signal is intended to migrate and chemically trigger processes in other external nucleic acid circuits. Such a stack data structure may eventually be embedded, for example, in an in vivo context to store messenger RNAs and reverse the temporal order of a translational response, among other applications.

Operation of the DNA stack data structure is as follows. To begin recording, a *start* (*s*) ssDNA strand is initially present in solution (Fig. 1d). *Start* defines the beginning of a stack complex. Then a *push* (*p*) ssDNA is added, which hybridises with *start* on the 28nt *A* domain leaving a 28nt single-stranded overhang *BC* at the 3′ of *push*. After the *start-push* reaction equilibrates (typical 30 min wait time), the system is subject to a manual mechanical washing step (see below). Following that, a hairpin signal strand (*X* or *Y*; Fig. 1e) is added to hybridise with *start-push* (*sp*) complexes on the exposed *BC* overhang. After a further wait and manual washing step, *push* is again added to hybridise with *start-push*-signal complexes in solution on the exposed *A* domain of the last signal. Repeating this stepwise process leads to elongating stack "polymers" that have nicks on alternate DNA backbones at 28 bp intervals (similar to DNA polymers in HCR[18]). As signals *X* and *Y* have identical hybridisation domains and hold different information only in their hairpin loops, the recording process can in principle record any arbitrary sequence of *X*'s and *Y*'s. Additionally, the signal set (alphabet) could be increased beyond binary.

Retrieval (popping) of the last signal recorded on the stack is carried out by adding *read* (*r*) strands to solution. *Read* hybridises the single-stranded overhang *A* at 5′ of the last signal recorded. It uses this domain as a long toehold and initiates strand displacement of the incumbent *push* strand hybridised to the signal. *Read* and signal (*X* or *Y*) form a blunt-ended double stranded product that irreversibly detaches from the end of the stack polymer. After a further washing step, *pop* (*q*) can be added to solution, which similarly removes the ultimate *push* strand by strand displacement. *Read* and *pop* can be alternated in this way to truncate the stack all the way back to the initial *start* strand. Figure 1f shows an example in vitro reaction sequence, and how this corresponds to operations applied to an in silico stack in computer programming.

The *push* strand is introduced between signals to reduce runaway polymerisation reactions, which would be more prevalent if two signals directly hybridised together. Another key strategy to reduce off-target reactions is the washing step referred to above ("W" symbols on Fig. 1d). Rather than floating free in solution, stack complexes are assembled attached to streptavidin coated beads. Before the stack reaction is started, sepharose beads anchor biotinylated *linker* (*k*) strands. *Start* then hybridises with *linker* strands, localising the majority of nucleating stack complexes to the beads (Fig. 1h). After each reaction stage completes, the system is washed by centrifuging and pouring out the supernatant solution, ideally leaving only beads with the on-target DNA stack polymers attached. Removal of the supernatant reduces interference with the next reaction stage by purging excess unreacted species and/or off-target species not attached to beads. The solution volume is replenished by addition of the next ssDNA strand. At the end of the process, all stack complexes tethered to beads can be released for analysis by addition of a *releaser* (*z*) strand. *Releaser* displaces *start* from *linker*, ejecting an entire stack complex attached to a bead into supernatant (Fig. 1h).

Important to emphasise is that the above account is a "single molecule" operational description of the DNA stack. It is strictly true when system volume is scaled down such that exactly one copy of each added species is present at the concentrations used. However, the bulk experimental scenario is far from single molecule: in a typical 20 μl volume, stack species are present in $10^{10}$–$10^{12}$ copies at nanomolar concentrations. Synchronisation of all stacks in solution is a central issue, as polymerisation processes can quickly generate a diversity of species (including potential ring polymers). The design choices mentioned above—i.e., irreversible reactions, reactions left sufficient time to fully equilibrate, and a periodic washing procedure to purge off-target species—attempt to enforce the synchronised assembly of stack species on the sepharose beads. Note also that single strands, and

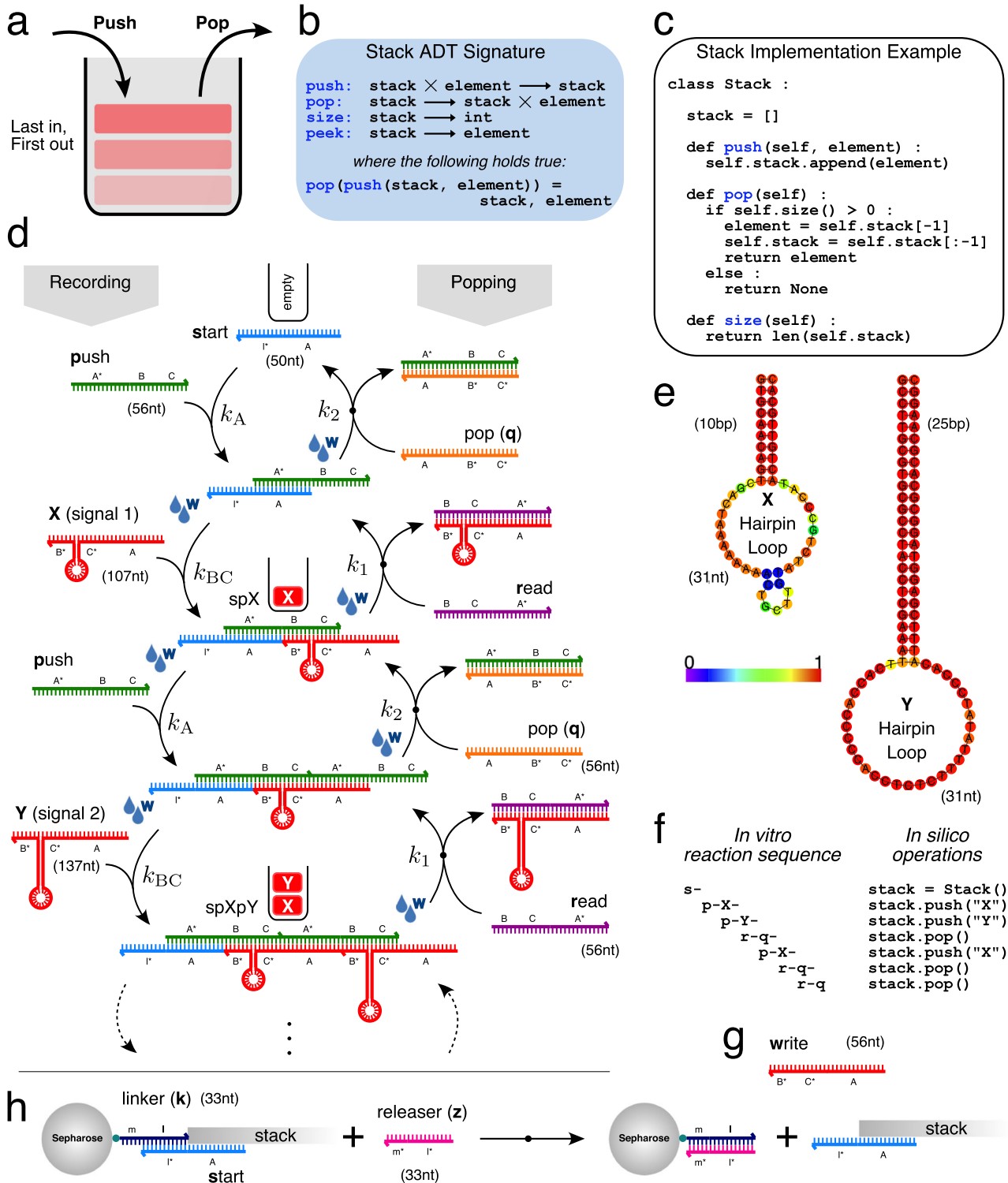

**Fig. 1 A last-in first-out stack data structure implemented in vitro with nine short ssDNA strands and operated at room temperature. a** Principle of a last-in first-out data structure; the last data item "pushed" is the first one "popped". **b** Formal specification of a stack as an Abstract Data Type. **c** A minimal stack implementation in Python (omitting `peek()`). **d** DNA reaction network implementation of a stack recording two signal types, *X* and *Y*. Hybridisation reactions result in polymerisation and recording of signals *X* and *Y* (left side). Conversely, strand displacement reactions (arrows with black dots) reverse polymerisation to retrieve signals in the order opposite to which they were recorded (right side). Domains marked as * are reverse complements, *W* denotes washing step (see text). **e** Secondary structure prediction of hairpin loops distinguishing *X* and *Y* signals (ViennaRNA: colour bar indicates base pairing probability). **f** Correspondence between in vitro DNA stack reaction sequence and *in silico* Python stack function calls `push()` and `pop()`. **g** Linear signal *w* (no hairpin loop) used to create a DNA stack system amenable to PAGE analysis and model fitting (see text). **h** Hybridisation/strand displacement method of tethering/releasing stack complexes to/from sepharose beads during washing procedure.

not strand complexes as in Qian et al.[17], are added to operate the stack. The monomer strand sequences were designed by a custom genetic algorithm[19] (Supplementary Note 1), and a first version of the strands was reported in ref. [20].

Using a computational model calibrated to experiments, below we report how performance of the DNA stack generally increases with increasing efficiency of washing steps between successive reaction stages. We also quantify how the performance of the device depends on the history of stack operations under inefficient washing, and suggest ways to overcome current limitations.

## Results

**DNA stack with a single signal type.** We first implemented the DNA stack with a linear signal strand (*write*) that has no secondary structure (Fig. 1g). *Write* created stack polymers similar to dsDNA duplexes, but with periodic nicks every 28 bp on alternate backbones and flanking ssDNA overhangs. These "duplex like" stack polymers were found to have two desirable characteristics: (i) they resolved to sharp, clearly separated bands on 10% polyacrylamide gel and (ii) they exhibited a predictable electrophoretic mobility whereby gel running base pairs were linearly proportional to the number of nucleotides in the stack complex (Supplementary Note 7). By contrast, stack complexes including signals *X* and/or *Y* with hairpin loops did not possess either of these features. For these reasons, we first operated the DNA stack with the linear *write* signal as a simplified system to get a handle on the reaction mechanisms of the polymer chemistry. A stack with a single *write* signal type is essentially a resettable counter, but from a thermodynamics and kinetics point of view, the underlying reactions are similar to the full stack system with two signal types because *write* strands bind by the same *A* and *BC* domains as the hairpin *X* and *Y* strands do. The washing procedure is also identical for both systems. Figure 2a–c show PAGE results of recording three *write* signals on the DNA stack and then popping the signals.

As can be seen, in addition to the target band at each reaction stage (blue arrows, Fig. 2a–c), multiple off-target bands also exist. To understand the factors leading to multiple off-target bands during stack operation, we developed a rule-based chemical kinetics model of the DNA chemistry and washing process (Supplementary Note 9). The five main reaction rules of the full 20-rule model are detailed in Fig. 2d. Reaction rate constants of the DNA chemistry were reduced to two key constants $k_A$ and $k_{BC}$, the bi-molecular hybridisation rate constants of the *A* and *BC* domains respectively. UV absorbance measurements yielded $k_A \approx k_{BC} \approx 3 \times 10^4\,\text{M}^{-1}\,\text{s}^{-1}$ (Supplementary Note 8). The two free parameters of the model related to the mechanical washing process: $0 \le \mu \le 1$ represented the fraction of beads lost on each wash and $0 \le \phi_0 \le 1$ represented the fraction of supernatant species transferred through the wash (for the initial mass of beads) via non-specific binding to the streptavidin coated sepharose beads and/or entrapment in solution pockets between beads.

Our motivation for including a non-specific DNA-bead interaction was the observation that, when omitting such an interaction, the model could not reproduce off-target stacks longer than the target stack. Additionally, non-specific interactions between DNA and similar streptavidin-coated magnetic beads have been reported previously[21]. Bands migrating higher than the target stack are made by, e.g., a residual concentration of *push* strands non-specifically carrying through the wash, which in turn form various complexes in solution on addition of the subsequent *write* strand (*pw*, *wp*, *pwp*, *wpw* etc.). Any such complexes beginning with *write* are then further able to join the end of the bead-tethered stacks.

Washing parameters $\mu$ and $\phi_0$ were collectively fitted to the 17 PAGE experiments of Fig. 2a–c using an error measure based on the intensity ordering of the banding pattern in the polyacrylamide gel (Supplementary Note 9.2). We estimated that $\mu = 0.1$, $\phi_0 = 0.33$ for our sepharose bead washing protocol (Fig. 2e). Figure 2 shows that predictions of the parameterised reaction model (blue boxes) well reproduce the main features of the 3-record 3-pop experiments using the *write* strand.

**Model predictions of DNA stack with two signal types.** Next, we extrapolated the parameterised model to exhaustively predict the (ideal) operation of the DNA stack with two signal types *X* and *Y*, this time over longer sequences of recording and popping operations.

The model was subject to five operation sequences each containing 20 record operations and 20 pop operations in total (Supplementary Note 10.1). In every case, approximately equal numbers of *X* and *Y* signals were recorded, and the stack always finished empty (under ideal operation). Four sequences, denoted *seqN*, were periodic, with *N* records always followed by *N* pops for $N = 1, 5, 10, 20$. One sequence, denoted *seqR*, arranged the 20 record and 20 pop operations randomly. Additionally, we tested the model with three increasingly stringent washing procedures. Procedure W1 was our experimental protocol ($\mu = 0.1$, $\phi_0 = 0.33$), procedure W2 was approximately twice as efficient ($\mu = 0.05$, $\phi_0 = 0.15$) and W3 was approximately twice as efficient again ($\mu = 0.02$, $\phi_0 = 0.05$).

For all operations sequences and all washing efficiencies, the model suggested that instead of detecting the most common stack species in the system, the most robust way to read out signals stored was to detect the majority signal popped into supernatant (*Xr* or *Yr*) following the addition of *read* strands (Supplementary Note 10.1). The model suggested that under imperfect washing, the stack population actually becomes de-synchronised from the target stack structure quickly (faster for W1 than for W3, as expected). However, despite this de-synchronisation of the stack population, the majority popped signal in supernatant remains correct for some time because the signals popped into supernatant derive from only the last signals stored on stacks in the population. That is, the stack population is simply required to have the majority of stacks terminating with the correct end signal, it is not required for all stacks in the population to be identical and synchronised (a much stricter condition). The fact that majority popped signal is the most robust system read out is indeed convenient, since the popped *Xr* and *Yr* complexes would be how the DNA stack is eventually linked to downstream nucleic acid circuits.

For reasons above, we evaluated the performance of a DNA stack as its "pop limit". In a run of recording and popping operations, "pop limit" is defined as the consecutive number of correct pop operations that can be performed before signals popped into supernatant (*Xr* or *Yr*) become indistinguishable, i.e., when $|[Xr] - [Yr]| < 10\,\text{nM}$. Figure 3 performs a systematic investigation of how washing efficiency, strand concentrations, reaction wait times, and pipetting noise affect the pop limit performance of the two signal DNA stack model on operations sequences *seq5* and *seqR* (drawn in Fig. 3a). See Supplementary Note 10 for *seq1*, *seq10*, and *seq20* results.

As expected, the stack model generally predicts that as $\mu$ and $\phi_0$ diverge from the perfect washing scenario, the pop limit of the stack system reduces, sometimes drastically so in a phase-transition-type behaviour (heatmaps of Fig. 3b, d and Supplementary Note 10.2). As $\mu$ increases from 0, beads get washed out of the system more quickly and the stack is able to do less total operations before the bead population carrying stacks is entirely lost. Conversely, as $\phi_0$

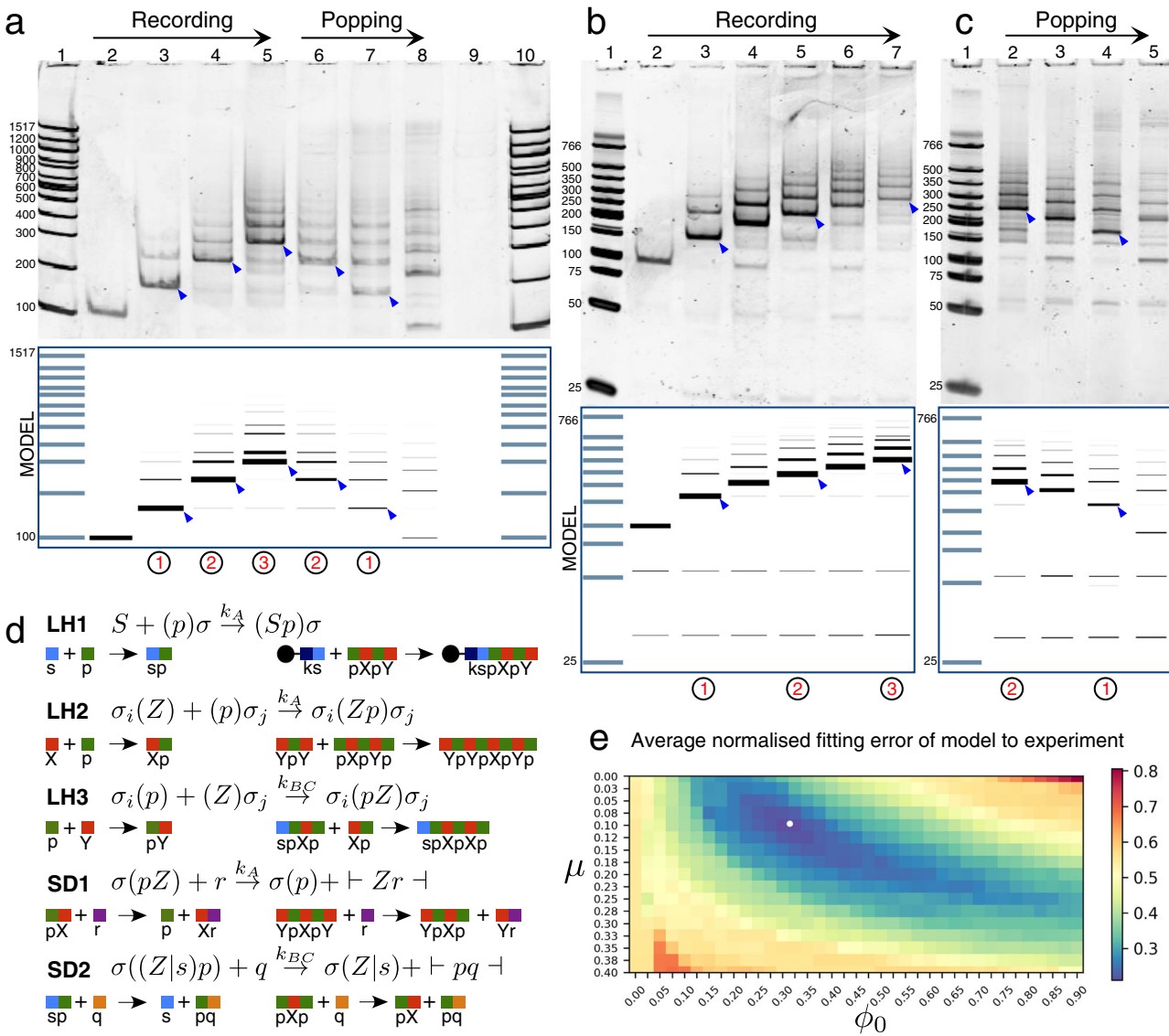

**Fig. 2 Experimental results and computational modelling of the DNA stack operated with a single signal type, *write*. a–c** Native PAGE results showing recording and popping up to three *write* signals. Blue boxes beneath show corresponding predictions of the mass action kinetics computational model. Black circled numbers denote the target number of *write* signals to be recorded in each lane; small blue arrows on gel images indicate the PAGE migration band corresponding to a stack complex with the target number of *write* signals. Ladder molecular weight markers in base pairs. Lane reaction sequences in Supplementary Note 3.1. All experiments were repeated independently three times. **d** Five principle reaction rules of the computational model. For clarity, each rule has two example reactions fitting the rule drawn underneath (rule notation in Supplementary Note 9). **e** Collective fitting of washing parameters $\mu$ and $\phi_0$ to experimental results in **a–c**, with white dot denoting lowest average error fitting at $\mu = 0.1$, $\phi_0 = 0.33$.

increases from 0, more excess supernatant species are non-specifically transferred through the wash to interfere with the next reaction stage, generally destroying synchronisation of the stack population which again limits total operations.

The exact pop limit that a non-perfect washing efficiency gives, however, depends on the exact sequence of operations applied to the DNA stack chemistry (i.e., performance is path dependent). Operation sequences with long stretches of records before pops (i.e., *seq20*) generally demand a more stringent washing procedure for good recall of stored signals on the stack. This is for two reasons: (i) stretches of record operations require more washes before popping operations are reached, during which the stack population diminishes if $\mu$ is not small; (ii) the stack population diversity increases with duration of a recording phase due to excess *push* and $X$ and $Y$ signals repeatedly carrying through the

wash if $\phi_0$ is not small. Conversely, popping operations tend to decrease species diversity (Supplementary Note 10.1).

For the five 40-operation sequences tested, the model stipulates that a strict washing efficiency in the range $0 < \mu < 0.03$, $0 < \phi_0 < 0.02$ is required for storage and correct popping of all 20 signals in all cases (largely dictated by *seq20*). It should be highlighted that certain pathological sequences like *seq1* (Supplementary Note 10.2) are insensitive to increasing $\phi_0$: in this case, the non-specific leak of supernatant species through the wash actually helps produce the correct read out at the next step. To some extent, this is also true for the *seq5* sequence used. We note that *seq5* sequences with different arrangements of $X$'s and $Y$'s can be constructed and may perform differently to the *seq5* sequence chosen. As a rough estimate, our estimated experimental washing efficiency W1 is predicted to achieve an average

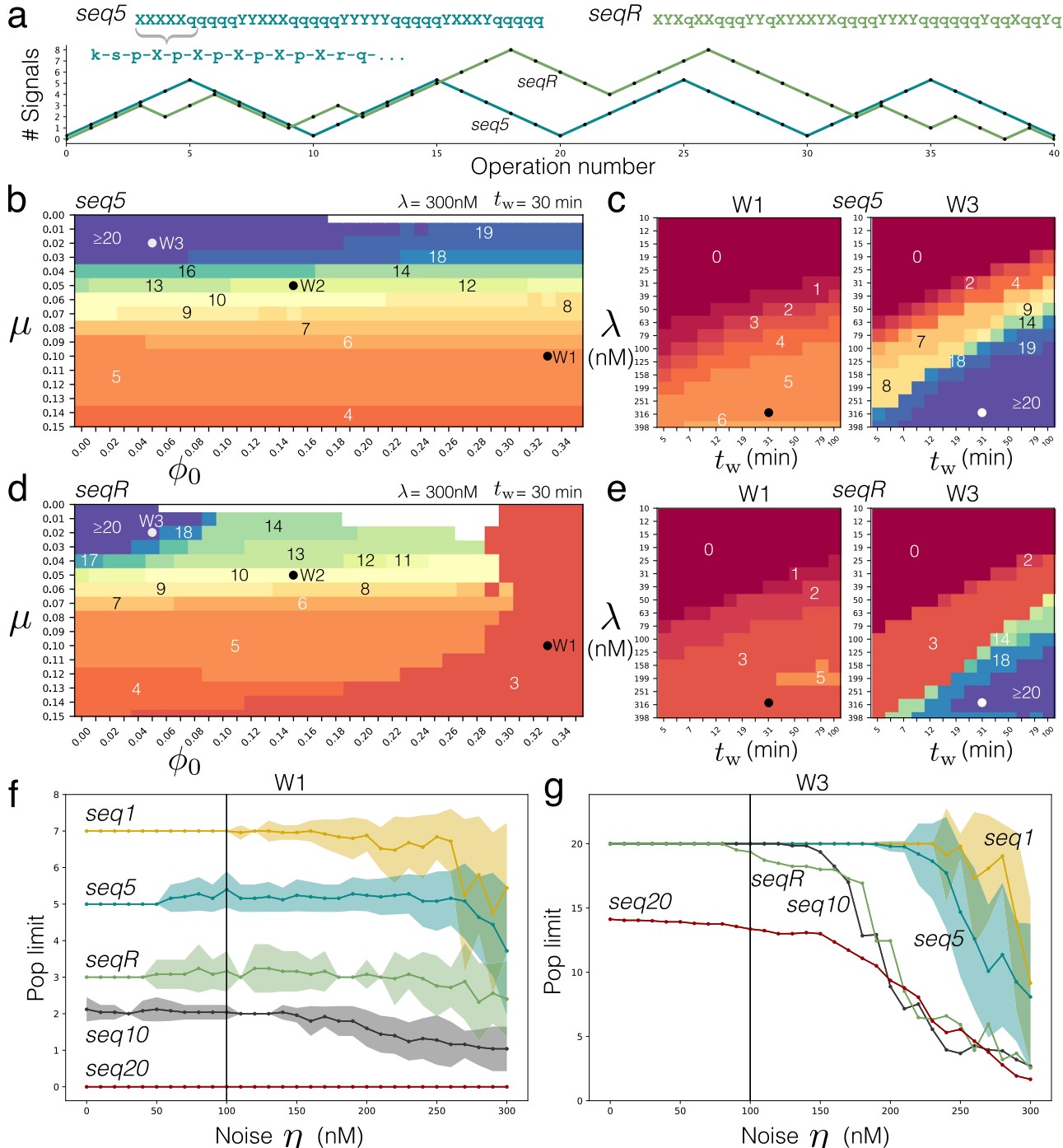

**Fig. 3 Model predictions of how the pop limit of a two signal type DNA stack is affected by different factors. a** Operations sequences *seq5* and *seqR* applied to model. Pyramid graph shows how stack elongates/truncates over time (total signals on stack) under each sequence (ideal operation). Heat maps **b** and **d** show how stack pop limit is sensitive to washing procedure efficiency for *seq5* and *seqR* respectively. Pop limit is denoted by overlaid numbers on heat map surface contours. Each heatmap square is a single simulation. Heat maps **c** and **e** show how pop limits vary at points *W1* and *W3* on (**b**) and (**d**), respectively, when strand concentrations $\lambda$ and reaction wait times $t_w$ deviate from standard conditions 300 nM and 30 min. Dots on heatmaps indicate position of standard conditions. Note the log scales. **f** How pop limit is affected by pipetting noise under *W1* washing. Pipetted concentration is drawn from uniform distribution 300 nM ± $\eta$. Average pop limit (lines) calculated from $n = 25$ independent stochastic simulations performed at each noise value, standard deviation shown as line shadows. **g** Same as **f** but using *W3* efficient washing: overlapping lines *seqR*, *seq10*, and *seq20* omit standard deviation shadows (drawn in Supplementary Note 10.4).

of 3.4 signals correctly popped per operations sequence, before output can no longer be discerned. This value generally agrees with our experimental observations of the two signal DNA stack system (see below).

Heatmaps in Fig. 3c, e show the predicted stack performance on *seq5* and *seqR*, for washing efficiencies W1 and W3, when concentrations and reaction wait times are changed from the standard $\gamma = 300$ nM and $t_w = 30$ min, respectively. See

Supplementary Note 10.3 for *seq1*, *seq10*, and *seq20*. Diagonal heatmap contours reflect the relationship that as wait times decrease, strand concentrations must increase to increase reaction rates and maintain performance. In our application, the concentrations of *linker* and *releaser* were always set to 200 nM while all other strands were set to $\gamma = 300$ nM. The heatmaps show $\gamma = 300$ nM is a reasonable operational setting. Decreasing concentrations below this point often leads to reactions taking too long to equilibrate, whereas increasing above 300nM no longer improves performance. Intuitively, high strand concentrations (e.g. $\gamma = 10\,\mu M$) could be expected to sharply degrade stack performance. This however is not predicted, at least under wash W2 (Supplementary Fig. 30). A partial explanation lies in the fact that species transferral fraction $\phi_n$ decreases exponentially from $\phi_0$ with successive washes; hence a large species excess only propagates non-specifically through 1 or 2 more washes than does a small one (Supplementary Note 9.5). The model suggests at $\gamma = 300$ nM, reaction wait times can be shortened from 30 min to increase device speed. The minimum wait time, however, is contingent on the operation sequence and washing efficiency. For W1 washing, the model predicts that wait times could be shortened to 10 min, whereas for W3 washing where the performance bar is higher, 20 min wait times could be used as a minimum.

Interestingly, the model forecasts that the two signal type stack with washing is quite insensitive to pipetting inaccuracies of strands (Fig. 3f, g). In fact, to negatively affect performance, strands must be added below the linker concentration of 200 nM (i.e., pipetting error must be $\eta > \pm 100$ nM in Fig. 3f, g). In this case, not all stacks on the beads receive the supplied strand, promoting increased stack population diversity and decreased synchronisation over time. Note that the effective linker concentration diminishes from 200 nM as more beads are lost. Pipetting above the linker concentration does not likely reduce the pop limit, for the reason that $\phi_n$ decreases exponentially over washes and larger excesses are only carried 1 more wash than are smaller excesses. Again, the exact level of pipetting noise required to reduce the stack pop limit is predicted to be contingent on the exact sequence of applied operations. Excessive noise downgrades performance at W3 washing efficiency (Fig. 3g) more than at W1 (Fig. 3f) simply because W3 leads to a more ordered chemistry where all stack complexes are semi-synchronised and noise has more potential to disorder the system.

Finally, increasing rate constants $k_A$ and $k_{BC}$ by revising the DNA stack strand sequences is predicted to extend existing device performance down to lower concentrations and shorter wait times, but not to improve general performance of the device (Supplementary Note 10.3).

**Experimental DNA stack with two signal types**. As a third step, we implemented the full two signal DNA stack using the hairpin loop *X* and *Y* strands of Fig. 1e. Recording and popping three signals gave the familiar "triangle" of increasing then decreasing stack sizes under capillary electrophoresis (Fig. 4a). In this case, only the dominant migration bands could be identified (blue arrows in figure) due to the unpredictable electrophoretic mobility of stack polymers that contain hairpin loops. Correct readout of signals popped from the DNA stack into the supernatant following addition of read strands could be obtained for a maximum of three signals—stored as YXX (Fig. 4c) or XXY (Fig. 4d) on the stack—under an optimised protocol where *read* was present at only 50 nM. Moreover, AFM imaging of a solution of stacks recording 3 *X* signals (Fig. 4f) revealed that peaks in the size distribution matched the expected nanometre lengths of stack polymers (Fig. 4g) with maximum density around the lengths of 2 and 3 signal stacks.

Curiously, repeated experiments found that supplying *read* at 300 nM could not achieve three signals correctly popped from the stack. These findings are in opposition to the stack chemistry model: for the experiments in Fig. 4, the model predicts *read* at 50 nM should lead to a maximum of only two signals retrievable, whereas supplying *read* at 300 nM should comfortably yield three signals. Short stack complexes on AFM micrographs are also not anticipated by the model when recording three signals (Supplementary Note 10.5). The model thus hints that additional unknown reaction processes are present in the DNA chemistry when the stack is operated with the *X* and *Y* hairpin signals, as opposed to being operated with just the linear *write* signal. We indeed confirmed this by finding significant synthesis by-products (ssDNA fragments) in the *X* and *Y* samples under denaturing PAGE (Supplementary Note 11). Signals *X* and *Y* are the longest synthetic oligos (at 107 nt and 137 nt, respectively), and thus are the most susceptible to synthesis errors such as truncated side products and oligos with internal deletions. HPLC and PAGE purification techniques by the manufacturer can only partially remove these undesired sequences[22]. Signal *Y* has the most impurities, consistent with it being the longest synthetic oligomer. This partially explains why AFM images of 3-signal stacks including *Y* (Supplementary Note 6) have more poorly resolved peaks in the object size distribution than when *Y* is absent (Fig. 4g).

Synthesis impurities in signals *X* and *Y* gave an unexpected opportunity to test and confirm the validity of the DNA stack chemistry model. Even if such impurities were eradicated, the model forecasts that DNA stack performance would still be limited to a pop limit of three or four signals at washing efficiency W1 (dependent on the exact operations sequence used). While minimisation of synthesis impurities is necessary for good stack performance, the key factor in raising pop limit in this system is a highly optimised washing protocol that avoids de-synchronisation of the stack population.

## Discussion

Our experimental DNA stack system constitutes proof-of-principle that a polymerising DNA chemistry can be used as a dynamic data structure to store two types of DNA signals in a last-in first-out order. Operation of the device was verified through PAGE although ultimately the stack would be operated and read out chemically by other interfacing nucleic acid circuits in the same solution.

The computational model indicates that significantly better performance is attainable with the existing DNA stack design if the experimental washing protocol is refined. In our experiments, streptavidin sepharose high performance beads were used without any blocking reagents. In future, non-specific interactions between beads and non-biotinylated ssDNA strands could be minimised by experimentally optimising the combination of bead type, blocking reagent and washing buffer. For example, different commercially available streptavidin magnetic coated beads could be tested in combination with BSA or PEG blocking solutions[21,23]. Saline-sodium citrate buffer (SSC) could be used at different concentrations and temperatures to control stringency during washing steps after hybridisation[21].

Computational modelling revealed that the in vitro DNA stack makes two important departures from a standard in silico stack data structure when the washing efficiency is not perfect. First, the stack state becomes path dependent on the actual operations sequence applied. That is to say, two different sequences of records and pops that should theoretically yield the same final stack state (i.e., *XqXYXq* and *XYXYqqYq*) actually yield different final states of the chemistry. This is due to the different specific

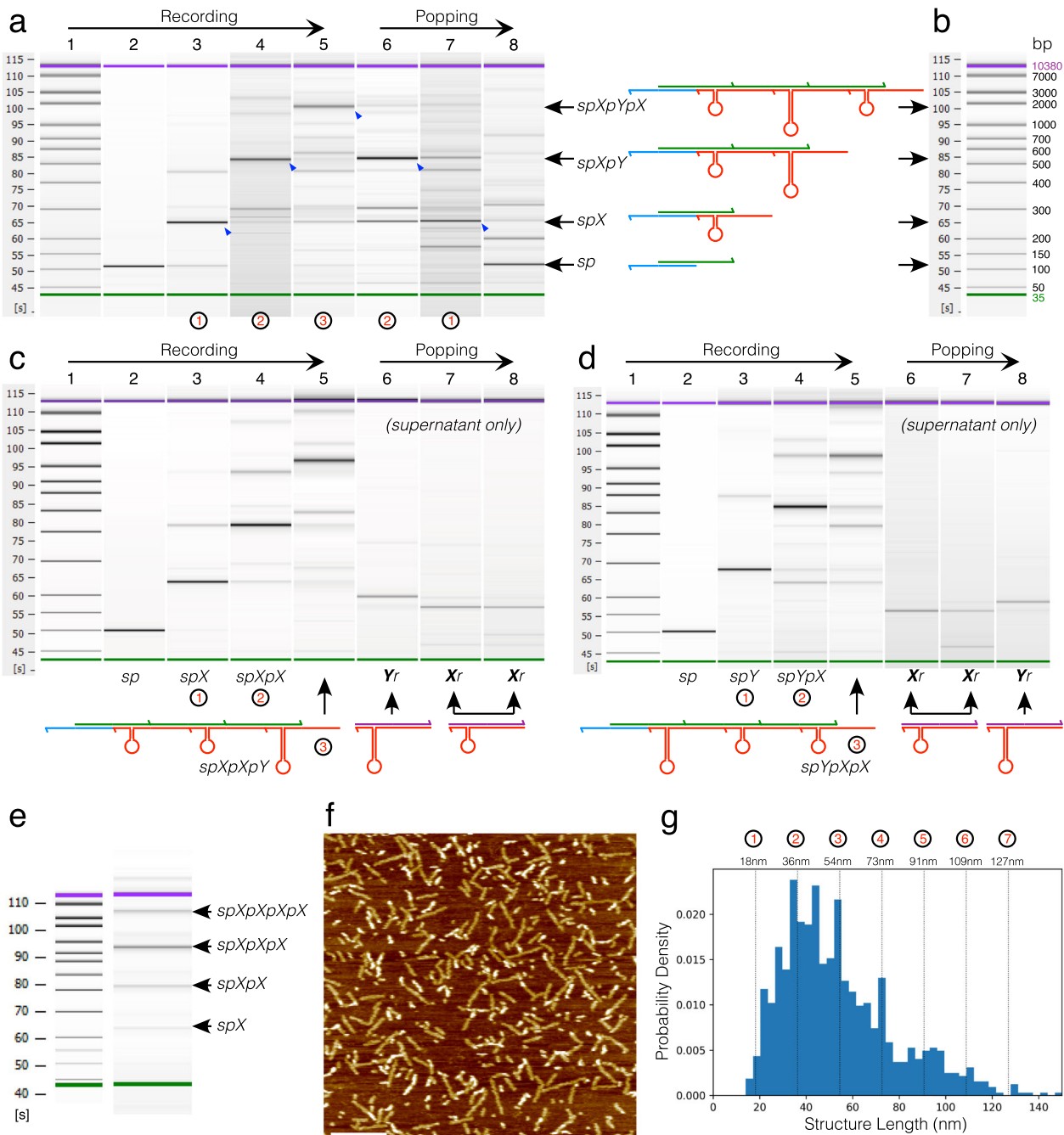

**Fig. 4 Experimental results for a DNA stack operated with two signal types *X* and *Y*. a** Capillary electrophoresis demonstrating elongation of stack complexes as signals *X*, *Y*, then *X* are recorded (lanes 2–5), followed by truncation of stack complexes during popping (lanes 6–8). Black circled numbers denote the target number of signals recorded in each lane. Blue arrows on virtual gel image denote migration band of target stack complex in each lane. **b** Conversion of capillary migration time ladder in seconds to approximate base pairs. **c**, **d** Recording signals on the stack data structure (lanes 2–5) and then analysis of only supernatant solution after *read* operations (lanes 6–8) revealing the double stranded signal complexes *Xr* and *Yr* popped off stacks. **c** *X*, *X* then *Y* is recorded, resulting in complexes *Yr*, *Xr* then *Xr* being popped into supernatant in reverse order. **d** *Y*, *X* then *X* is recorded, resulting in complexes *Xr*, *Xr* then *Yr* being popped. Lane reaction sequences in Supplementary Note 3.2. **e** Capillary electrophoresis of a stack sample where stacks are assembled with three *X* signals. **f** AFM micrograph of sample in **e**, scale bar 200 nm. **g** Distribution of stack assembly lengths on AFM micrograph (data from manual curation of 1022 objects in AFM field) showing discrete peaks at lengths corresponding to 2–7 signals. See Supplementary Note 6 for AFM of recording XYX and XXY. All experiments were repeated independently three times.

ways in which the stack polymer population becomes (irreversibly) de-synchronised via non-specific species transfer through the wash in each case. The result is that the history of operations performed on the DNA stack actually dictates whether future information can be recorded and popped or not. Path dependent performance lessens as washing efficiency improves, disappearing

at the singularity of perfect washing. The second departure from in silico operation is that the existing DNA design can perform only a finite number of operations. This is partly due to de-synchronisation of the stack population above, but also due to bead loss and leakage strand displacement reactions[24]. After the initial attachment of *start* to *linker* on beads, new stack polymers

cannot form in the chemistry and bead loss over time depletes the viable population.

The comparison of *in silico* and in vitro implementations is instructive to suggest more major design improvements. To make the DNA stack less sensitive to non-specific species carry over during washing, added strands could be engineered to only react with nucleating stacks on beads and not with each other in solution. For example, Qian et al.[17] accomplished this by using toeholds sequestered in added strand complexes, which only became exposed when the complex reacted with (and elongated) a stack polymer. Reactions creating side products in solution enhance stack population de-synchronisation and generally add to the complexity of the chemistry. But, when each possible reaction always requires a stack as one reactant, adding excess monomer only becomes significant if the excess propagates a few washes to the next time the same monomer type is added. Hence, in the latter case, $\phi_0$ has to be relatively large for interference to be introduced. Note that Qian et al.[17] relied on pre-prepared multi-stranded complexes to drive their system, which would not be natural outputs of third party nucleic acid circuits interfacing to the stack. It is an interesting design challenge to see if sequestered toeholds could be achieved with just secondary structure folds of single ssDNAs. An alternative route to diminish the effect of non-specific species carry over would be to modify the stack design to require the hybridisation of not one but three (or more) different *push* strands between signals (e.g., *s-p1-p2-p3-X-p1-p2-p3-Y*...). At the cost of operating speed, this simple strategy would increase the number of washes a species would have to be non-specifically carried across, before being able to interfere again. Moreover, this opens up for the possibility to operate several independent stacks in parallel in solution.

Leakage strand displacement reactions causing stack breakage and data loss over long time scales could be minimised by simple strategies, such as tethering individual stacks at separation distances that physically prohibit stack–stack invasions (for example on DNA origami surfaces[25] for small stacks), or via the use of GC pairs and/or mismatches at fray locations[26,24]. Note that such leakage reactions were not included in the chemistry model, and hence performance predictions at high operation numbers in Fig. 3 should be viewed as best estimates. Note also that the chemistry model assumed two-part hybridisation kinetics of DNA strands and eventual 100% completion of reactions. In reality, kinetic traps on the way to domain hybridisation could have led to <100% reaction completion at each stage, although this was difficult to capture in a systematic way.

Finally, the most challenging aspect is to create a stack that operates autonomously of a mechanical washing step. This would allow the device to be coupled with external nucleic acid circuits in an in vivo context, for example. The DNA stack exploits polymerisation to create a dynamic device without a fixed storage limit. In turn, however, this implies combinatoric species and reaction possibilities that must be kept in check by some form of population synchronisation. The stack can be operated without a mechanical washing step (Supplementary Note 4), but correct operation in this case is critically dependent on exact stoichiometric mixing of strands: a hard feat to achieve in the lab, let alone by nucleic acid reactions that are potentially interfacing with the stack. A potential solution could be to explore a DNA stack design based on amorphous computing principles[27,28], where the washing step is chemically mediated and intrinsic to the system. Under this approach, stack polymers would react directly with each other (and not just with added monomers) to implement a decentralised decision (population protocol) about which stack polymer is in majority (and thus should be the target polymer) at each reaction stage. As part of this, reactions could also actively amplify the target species at each stage, which is an

important aspect missing in the current design. If viable, this approach could also correct for the effect of zero toehold strand displacement breaking stacks over longer timescales. A population protocol for deciding approximate majority[29] of two species has been implemented in DNA[30]. However, it remains an interesting open question to see if such an approach can be scaled up to calculate, in reasonable time, the majority species in a system composed of many different sized stack complexes.

## Methods

**DNA stack assembly protocol.** HPLC purified DNA oligonucleotides were supplied by Eurogentec and resuspended in UltraPure™ water (Thermo Fisher Scientific) to give a stock solution of 100 μM and stored at −20 °C. Assembly reactions were performed in a buffer solution containing 8 mM magnesium acetate (Sigma Aldrich), 5 mM UltraPure™ Tris-HCl pH 7.5 (Thermo Fisher Scientific) and 1 mM UltraPure™ EDTA pH 8.0 (Thermo Fisher Scientific).

Washing steps were performed with Streptavidin Sepharose High Performance beads (GE Healthcare) following supplier's protocol. The streptavidin sepharose beads (100 μl) were incubated with 100 μl of the *linker* biotinylated single stranded DNA oligonucleotide (at 200 nM) for 30 min at room temperature with an Eppendorf Comfort Thermomixer at 300 rpm. Beads with the immobilised ligand were added to 1.5 ml centrifuge tubes, washed with filtered 1× Tris-EDTA (TE) and centrifuged for 5 s. Then the supernatant was carefully poured out (eliminating the excess of unattached strands) prior to the addition of the *start* DNA strand. The reaction mix with *start* added was incubated at room temperature and 300 rpm for 30 min, washed and centrifuged as above. The latter protocol (strand addition, incubation, and washing) was repeated for all subsequent strands.

The stack assembly was analyzed by polyacrylamide gel electrophoresis (PAGE) and Agilent 2100 Bioanalyzer. In detail, 10 μl of each completed sample was mixed with 1 μl of BlueJuice Gel Loading Buffer (Thermo Fisher Scientific) and loaded into the polyacrylamide gel wells. Samples were run on 10% Novex™ TBE gel (Thermo Fisher Scientific) in 1× Tris borate EDTA (TBE) buffer at 200 V for 35 min. After staining with SYBR® Gold (Thermo Fisher Scientific) in 1× TBE for 5 min, the gels were visualised using Typhoon laser scanner and ImageQuant TL software (normal sensitivity and PMT 500 or 600 V; GE Healthcare Life Sciences). Low molecular weight ladder and 100 bp ladder (NEB) were used as molecular weight markers. Agilent High Sensitivity (HS) DNA kit was used to run the samples following the manufacturer's protocol.

**Imaging of DNA stacks using AFM.** Topographic height images of DNA stacks were acquired using the liquid-phase tapping mode of MFP-3D Stand-Alone AFM (Oxford Instruments—Asylum research, Santa Barbara, CA, USA). Subsequent to the self-assembly step in the 0.2 ml tubes, the samples were diluted by a factor of four in the adsorption buffer (40 mM Tris, 12.5 mM MgCl₂, pH 8.5) and then 20 μl of the diluted sample was introduced onto the freshly cleaved mica substrate, fixed over a 150 μl-volume custom-made liquid cell. The adsorption step proceeded for 20 min and then 130 μl of the adsorption buffer (40 mM Tris, 12.5 mM MgCl₂, pH 8.5) was added, to cover the whole mica sheet with a ball-layer of buffer over the liquid cell. The imaging parameters used during the AFM imaging were: scan angle: 0°, scan rate: 1 Hz, AFM cantilever Olympus Biolever (BL-AC40TS-C2) with resonant frequency of 28 kHz in solution. Several images were acquired at different regions on the surface with minimum distance of 100 μm separation with respect to each other. The images were further analysed and processed using Igor Pro 6.37 A and ImageJ 1.52a.

**UV absorbance measurements.** Estimates of hybridisation rate constants were obtained by first preparing a 10 μM solution in water from 100 μM stock solution (resuspended oligos from Eurogentec) for each of the following strands: *push*, *pop*, *start*, *write*, and *p-glow*. Each 10 μM solution had its DNA mass concentration quantified (ng/μl, ten measurements) using NanoDrop One/OneC spectrophotometer (Thermo Scientific), selecting the option that takes into account sequence composition when calculating the extinction coefficient. Molarity of each solution was obtained using software available on line. Absorbance monitoring was carried out at 260 nm at constant 25 °C (averaging time 0.500, spectrophotometer band 2.000, Multicell Peltier UV–Vis, Agilent Cary 3500). Reaction mixes were prepared in 5 mM Tris-HCl, 1 mM EDTA, and 8 mM MgCl₂ buffer. Strands were mixed in pairs *push-pop*, *start-push*, and *write-pglow* such that, on mixing, the initial concentration of each strand was approximately 1 μM in a final volume of 70 μl. Before mixing, one of the strand pairs was prepared at elevated concentration in a cuvette volume of approx 62.5 μl to monitor prior absorbance. After an incubation of 5 min, the second strand was added to the cuvette, such that both strands became approximately 1 μM equimolar and the solution mixed via hand pipetting for 3 s. Afterward, absorbance was measured for a further 25 min (total incubation time: 30 min). Six independent kinetic measurements were recorded for each strand pair.

**Computational simulations**. Stochastic simulations of the DNA stack chemistry were performed in a volume of 0.15 pl using the Gillespie Direct SSA algorithm[31]. The stocal Python package (https://github.com/harfel/stocal) was used to run simulations, because it features just-in-time enumeration of next possible reactions to mitigate the state space explosion problem in the DNA polymer chemistry. Heatmap figures were computed using High Performance Computing resources at Newcastle University to run perfectly parallel simulations.

**Reporting summary**. Further information on research design is available in the Nature Research Reporting Summary linked to this article.

## Data availability

The authors declare that data supporting the findings of this study are available within the paper and its Supplementary Information files. The data set of polyacrylamide gels supporting Supplementary Note 7 have been deposited in the Zenodo database under accession code https://doi.org/10.5281/zenodo.5060760.

## Code availability

Open source Python simulation code of the DNA stack chemistry is available at https://bitbucket.org/engineering-data-structure-organoids/dnastack, released under an MIT licence. Installation and usage instructions are available at https://dnastack.readthedocs.io.

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

## Acknowledgements

We would like to thank Jaume Bacardit for assistance with the School of Computing HPC cluster at Newcastle University. This work is supported by EPSRC Grant EP/N031962/1 and a Royal Academy of Engineering Chair in Emerging Technologies (to N.K.).

## Author contributions

Conceptualisation: N.K. and H.F. Methodology: B.S.E., A.L., E.T., N.K., H.F., A.F.A.O., M.C. Investigation: A.L., B.S.E., E.T., A.F.A.O. Software: B.S.E. Writing—original draft: B.S.E. Writing—review and editing: B.S.E., A.L., E.T., A.F.A.O., M.C., H.F., N.K. Supervision: N.K., H.F., M.C., E.T. Project administration: N.K. Funding: N.K.

## Competing interests

The authors declare no competing interests.
