## [Peer Review File · Nature Communications]

Reviewers' Comments:

Reviewer #1:

Remarks to the Author:

Recommendation: Publication premature at current form

Comments:

In this work, Lopiccolo et al. built a Last-In-First-Out stack data structure as a form of DNA polymer using single-stranded push, recording, pop and read strands. The stack data is attached to the Sepharose bead via streptavidin-biotin interaction, which enables applying series of operations by washing steps. The construction and read the stack was analyzed in native PAGE gel analysis. The authors also performed extensive modeling to find the relationship between the bulk operation accuracy and experimental parameters such as loss of beads and the amount of residual DNA strands per step.

The idea of using DNA polymer in the form of nicked DNA is not completely new but the novelty of this paper is in using strand binding and strand-displacement as a method to write and reading data. However, I have two major concerns over the work, which potentially prevent it from being published in Nature Communications.

First, I understand the analogy between the stacked data and assembled nicked DNA. However, there's a big difference in the reading step. The data gets consumed and destroyed. Unlike in vivo DNA memory bits, the stack data in this manuscript cannot be replicated and only can be produced by sequence of writing reactions which requires knowledge of the data. Therefore, the destructive reading step prevents me from calling it a good data structure. A non-destructive protocol such as a nanopore experiment could have been a better choice.

Second, the yield of adding a correct bit is low. The yield in this work would allow recording of up to 3-5 bits as modelled in figure 3. Also, the authors claim the incomplete washing step was a cause for the error. However, if that was the case, one should see the appearance of shorter bands in the gel. I'm not sure if the image was cut or if the band was colour labelled, I don't see such shorter bands. Instead, I see multimer bands at the higher molecular weight. For example, the multimer band in the 3rd lane of Figure 4c has a length of around 80. And this exactly matches with the target band at lane 4. Considering that the same hairpins are added during this step, it makes sense that the multimer band from spX which is potentially spX(pX) migrates at the same speed as the spXpX. Therefore, I believe the low yield is due to the formation of multimers, which could be optimized by engineering the sequence or the reaction condition.

Reviewer #2:

Remarks to the Author:

This paper presents a molecular implementation of a last-in-first-out data structure using DNA reaction networks. The paper nicely illustrates how to record signals with hybridization reactions (both linear and hairpin domains) and to retrieve the signals through toe-hold strand displacement to reverse the recording sequence. The authors articulate the importance of wash steps for the data stack kinetics implemented in DNA and discuss the limitations of their work (mostly revolving around the washing step and/or potential questions about scalability). Overall, this paper nicely builds upon prior work and highlights another potential application in the DNA computing field.

Minor comments: I would encourage the authors to discuss if they foresee the signal readout to be only performed through gel visualization or if higher-throughput methods could be used. It is also not obvious to me if this data structure could be repaired if it were to break. Furthermore, I am having difficulty in understanding the motivation for a last-in-first-out data structure for non-volatile DNA archival storage. Is this more suited for search or others where you would want to read the last stored signal more often?

Reviewer #3:

Remarks to the Author:

This manuscript describes the use of sequential strand addition then strand displacement reactions

to implement a last in-last out data stack. The analogy to data stacks is compelling, the experimental implementation combined with model is a strength, and the deep analysis using the model and the very complete consideration of many of the fine details and possibilities is very nice. My two main suggestions would be:

1) that the text and some of the results especially in the supplementals can become dense and complex to understand. This is ok and important to include the details. But perhaps making the earlier parts of the paper more accessible would help draw more readers to the paper and 'stick with it' longer.

As one example, the layout of figure 1b is perhaps more complex than it needs to be. The overall idea is relatively straightforward but the figure made it more confusing for me. Specifically, it made it seem like each tier was a continually occurring cycle, rather than the path progressing down the left side then back up the right side. Perhaps figure 1e's setup is more intuitive, and that arrangement could be used to depict the process in a more linear fashion as well. The attachment to the bead also helps provide some physical intuition as to what is happening. Figure S14 also provides a potentially simpler schematic form that could be used.

2) the 'messiness' of the reactions doesn't seem to bode well for the practical use of this approach. Even in the best wash conditions, a 20 deep data stack doesn't seem very practical. Including some discussion about this would help. Even if not practical, the work is interesting and could spur thoughts about how to make it practical or inspire other approaches that are scalable.

Other minor comments:

The use of the term 'polymerisation' in line 80-81 can be somewhat confusing as this is not exactly a true polymerization. As written it implied dNTP by dNTP synthesis of a ssDNA strand.

Line 80-99 seems redundant with lines 100-132. Lines 80-99 were probably harder to understand than lines 100-132 so perhaps merging those into lines 100-132 would help readability.

What is the completeness of each step? Would unhybridized push strands lead to truncated stacks that 'skipped' a step. Relatedly, the authors provide pretty complete discussion of how to mitigate carryover of strands; but what about inefficient/incomplete hybridization of each step leading to truncated strands? What controls this, and could this be mitigated somehow?

Is there a limit to the size of the hairpins? For example, at some point, the desired hairpin might not form if there is sufficient complexity in the sequence to create undesired intramolecular hybridization. Some speculation about this would be interesting.

In figure 2a, could the authors speculate as to how larger bands than expected are being generated even at the recording step 3, 4, and 5? It is not clear how multimerization could be occurring to lead to these non-specific products so early in the steps, especially with these nonspecific bands being relatively strong in intensity.

Could the issue of larger bands than expected be partially mitigated by using several different read/pop sequences instead of just one set? That way if there is carryover, strands would have to be carried over multiple steps to have an effect.

We thank all reviewers for their time and valuable comments. Our responses follow. Alterations and additions in the paper PDF appear as yellow highlighted text. No alterations were made to the Supplementary Material.

Reviewer 1: I understand the analogy between the stacked data and assembled nicked DNA. However, there's a big difference in the reading step. The data gets consumed and destroyed. Unlike in vivo DNA memory bits, the stack data in this manuscript cannot be replicated and only can be produced by sequence of writing reactions which requires knowledge of the data. Therefore, the destructive reading step prevents me from calling it a good data structure. A non-destructive protocol such as a nanopore experiment could have been a better choice.

Authors Response R1-1: We thank the reviewer for raising these queries as they help us to clarify our intentions both in the paper as well as in this response.

Our paper is the first demonstration of a molecular implementation for a stack *data structure*. In computer science a stack data structure provides a realisation of the formal “interface” to a set of standardised operations (called signatures in formal specifications) to organise data. In Figure 1 of the paper we now show an example of a formal specification for a stack data structure (Fig 1b) and its realisation (implementation) in a procedural programming language (Fig 1c). As it can be seen from the formal specification and its implementation, reading (i.e. using the pop operation) takes that data item out of the stack. That is, in the terminology used by the reviewer, this is a destructive operation. Our molecular implementation of the stack data structure therefore does exactly what it is supposed to do: this is not a bug bit rather a feature of this data structure.

Our purpose in this study is not to create a novel type of DNA archival memory that is read with sequencing techniques (e.g. nanopore). Rather, our aim is to create a structured memory device that can be interfaced with external nucleic acid processes in solution via a standardised molecular interface (the interface being the push() and pop() functions of the stack data structure signature in Fig 1b). Thus, our work is aimed at dynamic memory in a DNA computing context, not static memory in a DNA archiving context. Our molecular data structure device retrieves signals in the opposite order to which they are recorded, similar to a stack data structure in Computer Science. Data popped from the stack is removed from the stack by necessity, in order to go into solution and react with external downstream nucleic acid components, triggering a response. To re-iterate, the pop operation in our system is intended to remove data and form complexes which will, ultimately, trigger a chemical response.

Alterations to paper: The comment by the reviewer helped us realise that more motivation for our work is required in the paper from the outset, to avoid misconceptions. We have made the following changes:

- 1) To the introduction, we re-arranged text and added new paragraphs (**lines 67-89**) to better explain how our DNA stack device is intended to be used in a DNA computing context.
 - 3) We modified the abstract (**lines 14-18**) to the same effect.
-

3) We added two new panels (b) and (c) to Figure 1, explaining the Abstract Data Type of a stack data structure and an example Stack implementation in Python, respectively. A further new panel Figure 1(f) shows how the DNA chemical implementation of a stack correlates to the in-silico Python implementation of a stack data structure.

4) We changed the title of the paper, to “A Last-In First-Out Stack Data Structure Implemented in DNA”, using “Data Structure” instead of “Memory”.

Reviewer 1: Second, the yield of adding a correct bit is low. The yield in this work would allow recording of up to 3-5 bits as modelled in figure 3.

Authors Response R1-2: We thank again the reviewer for giving us the opportunity to further clarify this point. Our work is not about “recording” data but about *organising* data while making it accessible via a standardised interface (i.e. the data structure signature alluded to earlier, also called an Abstract Data Type or ADT – see new panel Fig 1.b). Our demonstration shows that we can indeed organise up to 5 data items in any order (presented as X,Y in our examples) using functions of the stack ADT.

Why is the distinction between recording and organising data important? Consider this example: X & Y could, e.g., represent “access keys” to a data base. The item recorded under the access key could be very long indeed (e.g. the entire mRNA for a functional protein). The access keys organised in the stack could be used to selectively release the recorded mRNA in any order the stack ADT supports. Thus, the importance of a data structure is not about how much data each item holds but rather the manner in which it is written and retrieved.

The current DNA stack design, operated in the current experimental conditions is indeed limited to recording and popping up to 5 data items (although -in principle- more items could be stacked but with more degradation of performance). This is so because the paper is intended as a proof of concept (as stated in the discussion). To surpass the limitations of the current system, in the paper we already: (1) suggest through modelling how the performance of the device (the pop limit) can be increased through a higher efficiency washing procedure, (2) suggest how to experimentally achieve a higher efficiency washing procedure and (3) suggest how the stack design can be modified to better cope with an inefficient washing procedure. Therefore, we believe that the paper is novel and already a valuable contribution to the literature in its current state.

Alterations to paper: : We feel that this comment does not require alterations to the paper beyond the clarifications we have made to address the reviewer’s earlier point.

Reviewer 1: Also, the authors claim the incomplete washing step was a cause for the error. However, if that was the case, one should see the appearance of shorter bands in the gel. I’m not sure if the image was cut or if the band was colour labelled, I don’t see such shorter bands. Instead, I see multimer bands at the higher molecular weight.

For example, the multimer band in the 3rd lane of Figure 4c has a length of around 80. And this exactly matches with the target band at lane 4. Considering that the same hairpins are added during this step, it makes sense that the multimer band from spX which is potentially spX(pX) migrates at the same speed as the spXpX. Therefore, I believe the low yield is due to

the formation of multimers, which could be optimized by engineering the sequence or the reaction condition.

Authors Response R1-3: We consider our claim -- that inefficient washing is the main factor reducing device performance -- to be sound. Below we explain how this can cause the results displayed in the gel images.

The reviewer is correct, multimer bands do reduce the yield of the target band. However, it is inefficient washing that causes these multimer bands in the first instance. Multimer bands (taken to mean bands of higher molecular weight than the target band) result from inefficient washing in a simple way: if push carries through and signal X is added, various complexes like pX, XpX, Xp, pXp, pXpX form in solution, which are then able to attach the stacks on the beads, forming multimer stacks (even at early reaction stages).

Again, the reviewer is correct when they say that inefficient washing should cause shorter bands in the gel, and these are actually present in Figure 2 of the paper, migrating at around 50bp on the NEB low molecular weight ladder. The shorter bands are excess of releaser (over the amount required to release the bead-bound stacks), plus a residual from the strand added before the wash where releaser was applied (non-specifically transferring through that wash). Indeed, as shown in Figure S13, we found that 50bp was the running bp of single strands like start, push, write etc.

We emphasise that the appearance of both shorter bands, and of multimer bands of higher molecular weight is well predicted by our computational model.

On the issue of gel image cropping, uncut images of all of our polyacrylamide gels and BioAnalyzer images were already supplied in the Supplementary Information sections S3-S5.

Finally, the reviewer states that multimer bands could be reduced by engineering the sequences or reaction condition. We indeed already performed genetic algorithm optimisation of all sequences, as described in the introduction (**line 163**). We also have worked hard on optimizing reaction conditions, which allowed us to arrive at the protocol detailed in the methods section.

Alterations to paper: **Lines 213-219** have been added to briefly clarify the origin of multimer bands.

Reviewer 2: Minor comments: I would encourage the authors to discuss if they foresee the signal readout to be only performed through gel visualization or if higher-throughput methods could be used.

Authors Response R2-1: In the paper, the DNA stack is read through PAGE visualisation, but this is simply to *verify* the device operates correctly, it is not the ultimate intended readout. Ultimately, the device is intended to be *chemically* read by other nucleic acid circuits in the same solution. High throughput sequencing is not applicable to our DNA stack memory paradigm, as this is not a DNA archiving memory system.

As an alternative to PAGE verification, Solid State Nanopore (SSNP) verification could also be an option, but for the moment we considered this a more complex option. SSNP verification could be achieved following a methodology similar to that of Chen K, Kong J, Zhu J, Ermann N, Predki P, Keyser UF. Digital Data Storage Using DNA Nanostructures and Solid-State Nanopores. Nano Letters. 2019;19(2):1210-5. Namely, after releasing from the sepharose beads, the DNA stack in solution could be arrayed over a linear DNA carrier by hybridisation between a toehold array on the carrier and the start strand of the DNA stack. In this way, the DNA stack size could be determined by the average amplitude of a sequence SSNP current peaks generated by nanopore transit of the DNA stack-loaded carrier. Such current amplitude should reflect the molecular weight of the DNA stack, in analogy with its GE migration distance.

Alterations to paper: 1) The abstract and introduction of the paper has been changed (**lines 74-89**) to emphasise that our DNA stack data structure is intended as a dynamic memory system to be used in a DNA computing context.

2) On the topic of readout, the following sentence has been added to the discussion (**lines 432-435**):

“Operation of the device was verified through PAGE although ultimately the stack would be operated and read out chemically by other interfacing nucleic acid circuits in the same solution.”

Reviewer 2: It is also not obvious to me if this data structure could be repaired if it were to break.

Authors Response R2-2: We thank the reviewer for raising this issue, as it has helped us make some additional clarifications in the paper.

Alterations to paper: The discussion now includes a new consolidated paragraph on stack breakage via zero toehold strand displacement, and ways to mitigate this (**lines 503-512**). Also, repair after breakage is now mentioned as a potential ability of a population-protocol based approach to designing the stack system (**lines 535-537**).

Reviewer 2: Furthermore, I am having difficulty in understanding the motivation for a last-in-first-out data structure for non-volatile DNA archival storage. Is this more suited for search or others where you would want to read the last stored signal more often?

Authors Response R2-3: As stated in R2-1 above, the DNA stack system is not intended for *archival* storage of data. It is intended to be used in a *dynamic* DNA computing context, directly interfacing with other external processes utilising nucleic acid strands.

Alterations to paper: See response R2-1.

Reviewer 3: My two main suggestions would be:

1) that the text and some of the results especially in the supplementals can become dense and complex to understand. This is ok and important to include the details. But perhaps making the earlier parts of the paper more accessible would help draw more readers to the paper and 'stick with it' longer.

Authors Response R3-1: The reviewer's comments are acknowledged.

Alterations to paper: As described in Authors Response R3-5 below, the stack system is now described in a more "friendly" way, focussing on operational details, and omitting more abstract technical sentences (**lines 90-164**).

Also, to improve general readability:

1) The following mathematical sentence was removed from the paper: "Note that, as number of washes n advances in the model, parameter μ is constant and bead mass B_n exponentially decays: $B_n = (1-\mu)^n$. Conversely, the transfer parameter ϕ_n at wash n exponentially decays from ϕ_0 with the bead loss: $\phi_n = \phi_0 B_n$."

2) The following technical sentence was removed from the paper: "The device proceeds via effectively irreversible reaction steps, but it is operationally reversible in the sense that a stack polymer holding signals can be truncated back to a previous state at the expense of releasing double stranded waste complexes into the environment."

3) "Isothermally" is changed to "at room temperature" in Fig 1 caption.

4) The following paragraph was removed from the discussion, to make it shorter and flow better: "Further minor modifications to the existing DNA stack could include: (i) detecting which of the popped X_r and Y_r complexes have the majority in supernatant at the end of an experiment by adding extra strands that form e.g. an approximate majority detector^{\cite{Chen2013}}; (ii) using shorter linear strands differentiated by overhangs of different lengths for signals X and Y , to impart more predictable electrophoretic mobility to two signal stacks and to increase monomer synthesis purity; (iii) modifying the design such that signals are only reactive with external nucleic acid circuits after they have been popped off the stack and are complexed with $read$; (iv) splitting long reaction protocols over days by freeze-thawing the stack solution (Supplementary S5), allowing higher numbers of operations to be performed."

Technical detail remains unchanged in the Supplementary Material, as this detail is required to replicate the study.

Reviewer 3: As one example, the layout of figure 1b is perhaps more complex than it needs to be. The overall idea is relatively straightforward, but the figure made it more confusing for me. Specifically, it made it seem like each tier was a continually occurring cycle, rather than the path progressing down the left side then back up the right side. Perhaps figure 1e's setup is more intuitive, and that arrangement could be used to depict the process in a more linear fashion as well. The attachment to the bead also helps provide some physical intuition as to what is happening. Figure S14 also provides a potentially simpler schematic form that could be used.

Authors Response R3-2: We feel that Figure 1b (now Figure 1d) is an accurate general portrayal of our DNA reaction system, for the following reasons.

(1) Crucially, *it is not always the case* that the operation path progresses down the left hand side (adding all signals), and then back up the right hand side (popping all signals) as the reviewer suggests. Signals can be added and popped from the stack (and then re-added and popped again etc.) in an *ad hoc* arrangement, as is done in the SeqR recording sequence in the paper. The diagram reflects this possibility. Also, in a solution with many of the different ssDNA strands present, cycles at every “tier” is exactly what *would* happen in the DNA chemistry.

(2) Attachment to beads is not depicted in Figure 1d for the reason that this diagram is *general*. The DNA stack chemistry can also happen in free solution, as well as tethered to the beads, if operated in a certain way, such as when start is at higher concentration than linker and start carries through the wash. Additionally, the stack chemistry can be operated in a single tube without any beads (we indeed tried this: Figure S10, lanes 5,6,7), although the system is extremely sensitive to non-equimolar concentrations in this latter case.

In summary, if the diagram was altered to show only elongation of the stack to a certain number of signals, and then truncation back to 0 signals, it would not properly capture all the possible operation scenarios of the DNA stack.

Alterations to paper: To further help introduce the “logic” behind the chemistry depicted in figure 1d we have introduce a new panels (b) and (c) and (e) in Figure 1, where we show respectively the abstract data type of a stack data structure, an example implementation in Python and the correspondence between *in vitro* operation (a reaction sequence) and *in silico* operation (a series of function calls).

Reviewer 3: 2) the ‘messiness’ of the reactions doesn’t seem to bode well for the practical use of this approach. Even in the best wash conditions, a 20 deep data stack doesn’t seem very practical. Including some discussion about this would help. Even if not practical, the work is interesting and could spur thoughts about how to make it practical or inspire other approaches that are scalable.

Authors Response R3-3: The reviewer is correct: the current DNA stack design does require a highly efficient washing procedure in order to record 20+ signals (and then pop them) reliably. We must mention that (1) our discussion emphasises that the system is an important *proof of concept*, to serve as a future ideas base for subsequent designs. Also, (2) the discussion does already mention an important future route for potentially making the concept more practical; namely to move the stack design from a “single molecule design” to a “population-based protocol” which features stack-stack interactions as an error correcting mechanism.

Alterations to paper: A simple potential design change has been added to the discussion (**lines 494-502**) to make the DNA stack system more robust to inefficient washing. This addition was prompted by a later question by this reviewer (see Authors Response R3-9).

Reviewer 3: Other minor comments: The use of the term 'polymerisation' in line 80-81 can be somewhat confusing as this is not exactly a true polymerization. As written it implied dNTP by dNTP synthesis of a ssDNA strand.

Authors Response R3-4: The reviewer's comments are acknowledged.

Alterations to paper: This sentence has now been removed. We now also clarify what we mean by DNA polymers in the introduction on **line 78**.

Reviewer 3: Line 80-99 seems redundant with lines 100-132. Lines 80-99 were probably harder to understand than lines 100-132 so perhaps merging those into lines 100-132 would help readability.

Authors Response R3-5: We accept the reviewer's comment that there is substantial overlap in these sections.

Alterations to paper: In the introduction, these two sections are now merged to improve readability and we added an improved motivation of our approach (**lines 74-164**).

Reviewer 3: What is the completeness of each step? Would unhybridized push strands lead to truncated stacks that 'skipped' a step. Relatedly, the authors provide pretty complete discussion of how to mitigate carryover of strands; but what about inefficient/incomplete hybridization of each step leading to truncated strands? What controls this, and could this be mitigated somehow?

Authors Response R3-6: The standard reaction wait time used is 30 minutes, and reactions (both hybridisation and strand displacement) are considered to complete 100% in this time. The assumed 100% completion is based on two factors: (1) strands are always added in excess of the original linker strand, and so it cannot be the case that some stacks have insufficient strands to elongate (or be read) and (2) the reaction kinetics: an irreversible bimolecular reaction proceeding at a rate of $3 \times 10^4 \text{ M}^{-1} \text{ s}^{-1}$ (as measured for our system via UV absorbance) completes 100% within 30 minutes, for the nanomolar concentrations used.

Therefore, the problem of inefficient/incomplete hybridisation (or strand displacement) of steps is not relevant, for the 30 minute wait time used, and the carryover of strands is the main mechanism by which stack performance degrades.

When waiting times are less than 30 minutes, the issue of incomplete hybridisation (or strand displacement) is indeed relevant, but this issue is already fully dealt with in the paper: Figures 4c and 4e (and S25-S30 in Supporting Info) show that the model predicts decreasing waiting times lead to reduced stack performance due to incomplete reactions.

Alterations to paper: We feel that this comment does not require alterations to the paper.

Reviewer 3: Is there a limit to the size of the hairpins? For example, at some point, the desired hairpin might not form if there is sufficient complexity in the sequence to create undesired intramolecular hybridization. Some speculation about this would be interesting.

Authors Response R3-7: Engineering large hairpins (in terms of stem length and loop length) is not a requirement for our binary signal system. For interfacing the stack with other third party nucleic acid circuits, the hairpin loop only requires to be long enough to have a kissing loop interaction with another external nucleic acid strand. For example, in reference (18), we had used hairpin loops to bind 'reporter' strands decorated with gold nanoparticles for electron microscopy verification. The length of the hairpin stems only need be sufficiently different to differentiate signals on a gel readout: maximum size is not an issue here either.

It would only become relevant to work out how many different and gel-distinguishable hairpins there are if the system recorded more than 2 signal types.

Alterations to paper: We feel that this comment does not require alterations to the paper.

Reviewer 3: In figure 2a, could the authors speculate as to how larger bands than expected are being generated even at the recording step 3, 4, and 5? It is not clear how multimerization could be occurring to lead to these non-specific products so early in the steps, especially with these nonspecific bands being relatively strong in intensity.

Authors Response R3-8: With imperfect washing, multimerization can occur even at early steps through the following process. Take Lane 3 of Figure 2a. Here, reaction sequence s-p-w is performed. When push is added, some residual amount will be transferred in a non-specific way (i.e. not attached to stacks on the beads) through the wash. The push strands non-specifically transferred will react with the w strand added next, making various pw, wpw, wp, pwpw etc. polymer complexes in solution (sometimes even ring polymers). Complexes starting w in solution are further able to attach to stacks on the beads, making the many higher order bands. Shorter polymer complexes are easier to form in solution, due to less bi-molecular reaction steps required, and this is reflected in the fact that the higher order bands decrease in intensity. It's important to mention that the DNA chemistry model reproduces both the number and relative intensity of the high-order bands.

Alterations to paper: A paragraph has been added to the paper (**lines 213-219**), explaining how higher bands are generated, even at early steps.

Reviewer 3: Could the issue of larger bands than expected be partially mitigated by using several different read/pop sequences instead of just one set? That way if there is carryover, strands would have to be carried over multiple steps to have an effect.

Authors Response R3-9: The reviewer raises an interesting point here. Forcing strands to carry through more washes in order to have an effect could be implemented by having e.g. 3 different types of push strands between each signal. For example:

s-p1-p2-p3-X-p1-p2-p3-Y-p1-p2-p3-X...

Now, if p3 non-specifically transfers through wash before X is added: (1) it cannot hybridise to the “right” side of X (X-p3), as only p1 is able to do this, and p1 has long since been washed out of the system and (2) any p3 hybridising on the left of X in solution (p3-X) will not be able to hybridise to the end of stacks. Crucially, XpXpX complexes cannot form in solution. In order to interfere, each push would have to survive 3 washing steps (with probability of survival exponentially decreasing with each step). The same situation would exist for reading data from the stack (pop1, pop2 and pop3).

This design improvement for a non-perfect washing procedure however has the trade off that the device would be slower to operate (three pushes required between each signal, instead of one).

Alterations to paper: The above design improvement is now mentioned in the discussion, in the list of ways to improve the operating performance of the device (**lines 494-502**).

Reviewers' Comments:

Reviewer #1:

Remarks to the Author:

Thank you for addressing my comments on the previous version of the manuscript. I am happy with the answers from the authors and the changes made on the current version of the manuscript. I recommend publishing the manuscript as it is.

Reviewer #3:

Remarks to the Author:

The responses/revisions are satisfactory.

One minor comment for author consideration: regarding the assumption of 100% completion of each step due to the 30 min incubation and addition of push strands in excess...no reaction proceeds 100% to completion, especially these types of equilibrium and non-covalent reactions. Furthermore, there may be small hairpins or kinks, or hetero/homodimers that form transiently that block binding. So even if saturation/plateau is reached in a binding assay, it may not indicate 100% binding. Is this still possible in your system, did the absorbance measurements capture this possibility, and if not, this may suggest something to consider including in the manuscript discussion.

Reviewer 3: One minor comment for author consideration: regarding the assumption of 100% completion of each step due to the 30 min incubation and addition of push strands in excess...no reaction proceeds 100% to completion, especially these types of equilibrium and non-covalent reactions. Furthermore, there may be small hairpins or kinks, or hetero/homodimers that form transiently that block binding. So even if saturation/plateau is reached in a binding assay, it may not indicate 100% binding. Is this still possible in your system, did the absorbance measurements capture this possibility, and if not, this may suggest something to consider including in the manuscript discussion.

Authors Response: The reviewer's comment is valuable and is acknowledged.

In our DNA chemistry model, we make the simplifying assumption that all hybridisation (and strand displacement) reactions follow two-part kinetics and can be faithfully modelled as elementary bi-molecular reactions (an assumption widely used in the literature). Under this two-part kinetics assumption, the free energy changes of the 28bp A and BC domains hybridising to their complements (approx. -40 kcal/mol) suggest that the reactions will be heavily shifted toward products at equilibrium, with a vanishingly small reactant fraction remaining. This was the basis of our assumption that reactions were irreversible and completed 100%.

However, the reviewer raises the valid point that hybridisation reactions may *not* be two-part but may instead proceed via intermediate hybridisation stages (involving transient hairpins, bulges etc) before the full domain becomes completely bound. These intermediate stages may be reversible and lead to less than 100% reaction completion. We did consider this possibility at an early stage of the project, particularly considering that our hybridisation domains were 28 bases long. We ran 1 millisecond Multistrand simulations of the push-write hybridisation and found that the two strands were found in the target binding, but also in several clusters of intermediate microstates (involving various sized bulge loops in the hybridisation domain) at the end of the simulation time. However, computational tractability prohibited the simulation being run for longer than 1 millisecond and it was not clear if these kinetically trapped intermediate states would asymptotically resolve. The authors are not aware of modern computational strategies able to accurately predict the kinetics and equilibrium yield of nucleic acid hybridisation reactions. In this case then, the simplest hypothesis was to assume adherence to domain-level specification and hybridisation reactions completing to 100%.

The reviewer mentions that hetero/homodimer formation of reactants may also prohibit the target reaction from reaching 100%. However, we performed sequence design of the strands such that off-target dimer formation was minimised and so hetero/homodimer formation should not have been a factor.

In relation to our absorbance measurements, the ODE model we fitted implicitly assumed that hybridisation reactions completed to 100% at the absorbance baseline.

The reviewer's insightful comment suggests that we must more clearly emphasise in the paper and supplementary material that we are *assuming* two-part kinetics for DNA reactions and that this assumption is idealised.

Alterations to paper:

1) In the discussion of the paper, after the comment that leakage reactions were not present in the model, we added the additional model limitation: *“Note also that the chemistry model assumed two-part hybridisation kinetics of DNA strands and eventual 100% completion of reactions. In reality, kinetic traps on the way to domain hybridisation could have led to <100% reaction completion at each stage, although this was difficult to capture in a systematic way.”*

2) Supplementary Material section S8

Under equation 3, we added the sentence: *“Note that Equation 3 above assumes that the hybridisation reaction completes to 100% at equilibrium, which may not be strictly true if transient bind-states of the strands can kinetically block hybridisation.”*

3) Supplementary Material section S9

In the second paragraph, we acknowledge that modelling hybridisation and strand displacement as elementary bi-molecular reactions is an idealisation: *“DNA chemical reactions were modelled using bi-molecular mass action kinetics. DNA hybridisation and strand displacement reactions were assumed to follow a simplified two-part model that neglected intermediate binding states on the way to full domain hybridisation (although such intermediate states may have been present in the experimental system and could have led to less than 100% reaction yields at equilibrium).”*